

# The complete chloroplast genomes of three Betulaceae species: implications for molecular phylogeny and historical biogeography

Zhen Yang, Guixi Wang, Qinghua Ma, Wenxu Ma, Lisong Liang and Tiantian Zhao

Key Laboratory of Tree Breeding and Cultivation of the State Forestry and Grassland Administration, Research Institute of Forestry, Chinese Academy of Forestry, Beijing, China

Corresponding author
Tiantian Zhao,
zhaotian1984@163.com

## ABSTRACT

**Background**. Previous phylogenetic conclusions on the family Betulaceae were based on either morphological characters or traditional single loci, which may indicate some limitations. The chloroplast genome contains rich polymorphism information, which is very suitable for phylogenetic studies. Thus, we sequenced the chloroplast genome sequences of three Betulaceae species and performed multiple analyses to investigate the genome variation, resolve the phylogenetic relationships, and clarify the divergence history.

**Methods**. Chloroplast genomes were sequenced using the high-throughput sequencing. A comparative genomic analysis was conducted to examine the global genome variation and screen the hotspots. Three chloroplast partitions were used to reconstruct the phylogenetic relationships using Maximum Likelihood and Bayesian Inference approaches. Then, molecular dating and biogeographic inferences were conducted based on the whole chloroplast genome data.

**Results**. Betulaceae chloroplast genomes consisted of a small single-copy region and a large single copy region, and two copies of inverted repeat regions. Nine hotspots can be used as potential DNA barcodes for species delimitation. Phylogenies strongly supported the division of Betulaceae into two subfamilies: Coryloideae and Betuloideae. The phylogenetic position of *Ostryopsis davidiana* was controversial among different datasets. The divergence time between subfamily Coryloideae and Betuloideae was about 70.49 Mya, and all six extant genera were inferred to have diverged fully by the middle Oligocene. Betulaceae ancestors were probably originated from the ancient Laurasia.

**Discussions**. This research elucidates the potential of chloroplast genome sequences in the application of developing molecular markers, studying evolutionary relationships and historical dynamic of Betulaceae. It also reveals the advantages of using chloroplast genome data to illuminate those phylogenies that have not been well solved yet by traditional approaches in other plants.

## INTRODUCTION

The family Betulaceae in the order Fagales consist of approximately 100~150 species of trees and shrubs that distributed in the temperate zone of the Northern Hemisphere, with a few species spreading to South America and only one species (*Alnus glutinosa* (L.) Gaertn) occurring in Africa (*Kubitzki, Rohwer & Bittrich, 1993*). This family is well-defined to contain six genera, five of which (*Betula*, *Alnus*, *Corylus*, *Ostrya*, and *Carpinus*) display similar patterns of intercontinental disjunction between Eurasia and North America, whereas *Ostryopsis* is only endemic to China. The typical features of Betulaceae are their doubly serrate, stipulate leaves, small winged fruits or nuts associated with leafy husks, and catkins appear before leaves.

The monophyly of Betulaceae is supported by numerous synapomorphies, such as compound catkins (*Abbe, 1974*), pollen micromorphology (*Chen, 1991*), growth habitat (*Kikuzawa, 1982*), and embryology (*Xing, Chen & Lu, 1998*). However, the generic relationships within the family have subjected to various controversies. In previous studies, both morphological taxonomy and molecular phylogenies have generally recognized two main lineages in Betulaceae, treated either as two tribes (Coryleae and Betuleae) (*Bousquet, Strauss & Li, 1992*; *Crane, 1989*) or two subfamilies (Coryloideae and Betuloideae) (*Furlow, 1990*; *Bousquet, Strauss & Li, 1992*; *Chen, Manchester & Sun, 1999*; *Forest et al., 2005*). Meanwhile, some other taxonomists upgraded the two lineages as two families Corylaceae and Betulaceae sensu stricto (*Dahlgren, 1983*; *Hutchinson, 1967*). Recent treatments (*Xiang et al., 2014*; *Grimm & Renner, 2013*; *Soltis et al., 2011*), including the Angiosperm Phylogeny Group (*APG III, 2009*; *APG IV, 2016*), also have described the two lineages as subfamilies within an expanded Betulaceae: Betuloideae (*Betula* and *Alnus*) and Coryloideae (*Ostryopsis*, *Corylus*, *Ostrya*, and *Carpinus*). Nevertheless, all the above taxonomic and phylogenetic conclusions are inferred from unreliable and dynamic morphological features or DNA fragments with limited polymorphic information loci (e.g., *rbc* L, *mat* K, and ITS), which may inevitably bias the phylogenetic reference (*Philippe et al., 2011*). Especially, due to recent speciation and rapid diversification, the generic relationships within the subfamily Coryloideae are still phylogenetically and taxonomically difficult (*Forest et al., 2005*; *Yoo & Wen, 2002*; *Chen, Manchester & Sun, 1999*; *Kato et al., 1998*). Additionally, future studies on Betulaceae will pay more attention to species identification, population genetics, and biogeographic origin. All these studies rely on high-resolution molecular markers and robust phylogeny, but the limited and low-resolution DNA markers heavily inhibited the comprehensive evaluation of Betulaceae resources. Therefore, it is imperative to develop efficient molecular markers to resolve the current problems.

Chloroplast (cp) genome is one of the three sets of genetic systems (cytoblast, chloroplast, and mitochondrion) with different evolutionary histories and origins in higher plants. Generally, phylogenetic inferences using nuclear genomes are unrealistic for their costly situation and lack of enough genomic data (*Wang et al., 2014*; *Olsen et al., 2016*). Meanwhile, mitochondrial genomes are not suitable for phylogenetic studies of plants due to their slow evolutionary rate and rich in exogenous sequences (*Palmer & Herbon, 1988*). Compared to nuclear and mitochondrial genomes, cp genomes have independent

evolutionary routes and own the characteristics of uniparental inheritance, moderate rates of nucleotide substitutions, haploid status, and no homologous recombination (*Hansen et al., 2007*; *Shaw et al., 2005*). Correspondingly, these features of cp genomes make them particularly suitable for phylogenetic and biogeographic studies of plants (*Attigala et al., 2016*; *Walker, Zanis & Emery, 2014*; *Huang et al., 2014*). With the accumulation of angiosperm cp genomes, comparative genomics and phylogenomics of closely related cp genomes are very useful for grasping the genome evolution regarding structure variations, nucleotide substitutions, and gene losses (*Hu, Woeste & Zhao, 2017*; *Raman & Park, 2016*; *Barrett et al., 2016*). Meanwhile, lots of high-resolution genetic markers, such as intergenic spacer (IGS) fragments (*Liu et al., 2016*), simple sequence repeats (SSRs) (*Huang et al., 2014*), single nucleotide polymorphisms (SNPs) (*Li et al., 2014*), and repeated sequences (*Provan, Powell & Hollingsworth, 2001*) were identified across the cp genomes and applied for multi-aspect studies in different plant taxa.

Currently, the cp genomic resources of Betulaceae are fairly limited, and much less for some rare species from the genera *Corylus* and *Alnus*. Especially, no cp genome is available for the genus *Ostryopsis*. Here, we sequenced the complete cp genome sequences of three Chinese endemic Betulaceae species (*Ostryopsis davidiana*, *Corylus wangii*, and *Alnus cremastogyne*) that are narrowly distributed in limited regions and are poorly studied in previous research, then, comparative genomics and phylogenomics analyses were conducted by integrating previously published cp genomes from other taxa in Betulaceae. Our aims are to compare and characterize the cp genomes among selected species of Betulaceae; identify and screen molecular markers suitable for population genetics; reconstruct the intergeneric relationships of the six extant genera of Betulaceae; estimate the divergence time and biogeographic history of Betulaceae.

## MATERIALS & METHODS

### Plant materials, DNA isolation and sequencing

Fresh plant leaves of three Betulaceae species were harvested from their natural populations in China, including *Ostryopsis davidiana* from Chifeng, Neimengu, *Corylus wangii* from Weixi, Yunnan, and *Alnus cremastogyne* from Wuxi, Chongqing. Voucher specimens were stored in herbaria of Research Institute of Forestry, Chinese Academy of Forestry. Total genomic DNA was extracted from silica-dried leaves using a modified CTAB protocol (*Li et al., 2013*) and purified employing the Wizard DNA CleanUp System (Promega, Madison, WI, USA). DNA samples were fragmented randomly and then were sheared into 400–600 bp fragments through agarose gel electrophoresis. The paired-end libraries with 500 bp insert size were built using the Illumina PE DNA library kit, and then paired reads were sequenced with an Illumina HiSeq 4000-PE150.

### Chloroplast genome assembly and annotation

We used SPAdes 3.6.1 (*Bankevich et al., 2012*) to initially assemble the cp genomes under the '-careful' option with k-mer sizes of 21, 33, 55, 77 and 89. SPAdes contigs were further blasted against the *Corylus heterophylla* (KX822769) and *Alnus alnobetula* (MF136498) cp genomes using blastn with an e value cutoff of $1e^{-10}$ to filter out chloroplast-like contigs

(*Camacho et al., 2009*). Then, these chloroplast contigs were assembled using Sequencher v5.4 software. Finally, Geneious 8.1 (*Kearse1 et al., 2012*) was used to map all the reads onto the assembled chloroplast genome to verify the accuracy. Based on the reference sequence, the junctions among large single copy (LSC) region, two inverted repeat (IRa and IRb) regions, and small single copy (SSC) region were verified following the method of *Dong et al. (2014)*. Annotations of the three chloroplast genomes were performed using the online program DOGMA (*Wyman, Jansen & Boore, 2004*) with default parameters. Positions of introns, starts, and stops were checked by aligning with homologous genes of *Corylus heterophylla* (KX822769) and *Alnus alnobetula* (MF136498) cp genomes using MAFFT v7.0.0 (*Katoh & Standley, 2013*). In addition, annotations of transfer RNAs were further verified with tRNAscan-SE search server (*Schattner, Brooks & Lowe, 2005*). The cp genome map was plotted with Genome Vx software (*Conant & Wolfe, 2008*). The annotated cp genome sequences of *Ostryopsis davidiana*, *Corylus wangii*, and *Alnus cremastogyne* have been submitted to GenBank (accession numbers MH628451, MH628454, and MH628453).

## Comparative analysis and sequence divergence

In order to evaluate the sequence divergence of Betulaceae cp genomes, we randomly selected six of the available Betulaceae species (one representative for each of the six genera), including three cp genomes we reported here plus the cp genomes of *Carpinus tientaiensis* (KY174338), *Betula nana* (KX703002), and *Ostrya rehderiana* (KT454094). Based on previous studies, the contraction and expansion of IR regions could bring about the structure variation and length change of cp genomes (*Nazareno, Carlsen & Lohmann, 2015*; *Yang et al., 2018*). Thus, we performed a comparative analysis to test the variation in the IR/SC junctions among Betulaceae cp genomes. To assess rearrangement and substantial sequence divergence, we conducted a synteny analysis using the progressive Mauve aligner implemented in Mauve 2.3.1 (*Darling, Mau & Perna, 2010*) under default settings. To screen polymorphic hotspots that can be used as molecular markers to identify Betulaceae species, 79 shared protein-coding genes (PCG) and 121 intergenic spacer regions (IGS) of the six cp genomes were separately extracted. These homologous regions were aligned using MAFFT 7.0 and then adjusted manually with Se-Al 2.0 (*Rambaut, 1996*). Subsequently, the number of variable sites and aligned sequence length for each region was calculated using DnaSP 5.0 (*Librado & Rozas, 2009*), and the percentages of variable sites = (number of variable sites/aligned sequence length) $\times 100$.

## Repeated sequences and microsatellites

We employed the online REPuter software (*Kurtz et al., 2001*) to scan and visualize forward, reverse, complement, and palindromic structure with a minimum repeat size of 30 bp and edit distances of less than 3 bp. Tandem repeats were identified using the online software Tandem Repeats Finder 4.07 b (*Benson, 1999*), with the match, mismatch, and indel parameters separately set as 2, 7, 7. The minimum alignments score and maximum period size were assigned 70 and 500, respectively. Microsatellites or simple sequence repeats (SSRs) were predicted with Msatcommander 0.8.2 (*Faircloth, 2008*). We set the threshold for mono-, di-, tri-, tetra-, penta-, and hexa-nucleotide SSRs with ten, five, four, three, three, and three repeat units, respectively.

## Phylogenetic inference

In order to infer the intergeneric relationships within Betulaceae, eleven representative cp genome sequences from the six genera (*Betula*, *Alnus*, *Corylus*, *Carpinus*, and *Ostrya*) of Betulaceae were applied to construct phylogenetic trees, with two species from the genus *Juglans* (*Juglans regia* and *Juglans nigra*) selected as outgroup taxa. These cp genomes and GenBank accession numbers are listed in Table S1 . To evaluate the utility of different structural domains, phylogenies were inferred based on three datasets: (1) complete cp genome sequences (CPG); (2) protein-coding genes (PCG); (3) intergenic spacer regions (IGS). Each dataset was aligned using MAFFT 7.0 with default parameters and ambiguously aligned sites in all alignments were removed using Gblocks v.0.91b (*Talavera & Castresana, 2007*) with all gap positions allowed. Two different phylogenetic algorithms were employed in this analysis: maximum likelihood (ML) method and Bayesian inference (BI) method. We conducted the ML analysis using IQ-tree 1.6.3 (*Nguyen et al., 2015*) with 1,000 replicates of ultrafast bootstrapping (UFBoot) (*Minh, Nguyen & Haeseler, 2013*), 1,000 bootstrap replicates of the Shimodaira/Hasegawa approximate likelihood-ratio test (SH-aLRT) (*Guindon et al., 2010*). The best-fit model for each sequence partition was predicted by the built-in ModelFinder program (*Kalyaanamoorthy et al., 2017*) of IQ-tree under the Bayesian information criterion. TVM + F + R3, TVM + F + I, and GTR + F + R2 substitution models were selected for CPG, PCG, and IGS, respectively. BI analysis was performed using MrBayes 3.2.6 (*Ronquist et al., 2012*) under GTRGAMMA model, with four chains and two parallel runs. Each run was conducted until completion, and included 1,000,000 generations, with sampling every 100 generations. The first 25% of the trees were discarded as burn-in and the remaining trees were used for generating the consensus tree. The final trees and posterior probabilities were visualized with FigTree v1.4 (*Rambaut, 2012*).

## Molecular dating analysis

We performed a time-calibrated coalescent Bayesian analysis in BEAST 2.48 (*Bouckaert et al., 2014*) to estimate the divergence times of Betulaceae lineages at genus level. BEAST is a cross-platform program for Bayesian analysis of molecular sequences using Markov chain Monte Carlo (MCMC). It is entirely orientated towards rooted, time-measured phylogenies inferred using strict or relaxed molecular clock models. In this study, we estimated divergence times using a gamma-distributed rate variation, a proportion of invariant sites of heterogeneity model, and estimated base frequencies. An uncorrelated log-normal clock was applied with a Yule process speciation prior for branching rates. Two fossil constraints were used for calibration: (1) the crown age of the family Betulaceae was set to 69.95 Mya (SD = 2.0) and assigned a normal distribution (*Xiang et al., 2014*); (2) A prior for the calibration of the most recent common ancestor (MRCA) for the subfamily Coryloideae was included following a normal distribution with mean 48 Mya (SD = 0.5) (*Pigg, Manchester & Wehr, 2003*). We ran 500 million MCMC generations with a sampling frequency of 1,000 generations after a burn-in of 1%. The convergence of parameters was checked with Tracer v1.6 (*Rambaut et al., 2014*), confirming effective sample size (ESS) was

greater than 200. Maximum clade credibility (MCC) trees were computed after discarding 1% of the respective saved trees as burn-in.

### Ancestral area reconstruction

To grasp the biogeographical history of Betulaceae, we performed an ancestral area reconstruction. Six areas were designated based on the tectonic history of continents and the current distribution data of Betulaceae species: A, East Asia; B, Europe; C, North America; D, Central America; E, South America; F, North Africa. Based on the MCC tree obtained from BEAST, the Bayesian binary MCMC (BBM) method in RASP 4.0 (*Yu et al., 2015*) was used to reconstruct the ancestral areas of Betulaceae species. MCMC chains in the BBM analysis were run for 10 million generations with a sampling frequency of 100, discarding the first 1,000 generations as burn-in. The number of maximum areas was maintained at four.

## RESULTS

### Chloroplast genome sequencing and assembly

Using the Illumina HiSeq 4000-PE150 platform, we newly sequenced the cp genomes of three Betulaceae species (*Ostryopsis davidiana*, *Corylus wangii*, and *Alnus cremastogyne*). Overall, Illumina paired-end (2 × 150 bp) sequencing generated large datasets for each species, with 8,683,726 (*Ostryopsis davidiana*), 22,450,682 (*Corylus wangii*), and 27,361,376 (*Alnus cremastogyne*) paired-end reads mapped to the reference genome sequences, resulting 777×, 132×, and 785× coverage across the three cp genomes. The results indicated that the quality of cp genome sequencing and assembly was very high.

### Organization of Betulaceae chloroplast genome

The availability of three other complete cp genomes of Betulaceae species (*Carpinus tientaiensis*, KY174338; *Betula nana*, KX703002.1; *Ostrya rehderiana*, KT454094) provided an opportunity to compare the cp genome organization and sequence variation within this family. Organization of the Betulaceae cp genome was quite conserved; neither inversions nor translocations were observed in the analysis. The six cp genomes ranged from 159,286 base pairs (bp) (*Ostryopsis davidiana*) to 160,579 bp (*Betula nana*) in length. The six cp genomes displayed a circular quadripartite structure including two IR regions (ranging from 25,927 bp in *Ostrya rehderiana* to 26,185 bp in *Alnus cremastogyne*), the LSC region (ranging from 88,552 bp in *Ostrya rehderiana* to 89,493 bp in *Betula nana*), and the SSC region (18,588 in *Ostryopsis davidiana* to 19,094 bp in *Alnus cremastogyne*) (Table 1, Fig. 1). Differences in genome size mainly resulted from the length variation of the SC regions, with minor discrepancies observed among IR regions. The GC content was roughly identical among the six cp genomes, ranging from 36.07 to 36.68%.

Each of the six Betulaceae cp genomes encoded 131 genes, of which 113 genes were unique, and 18 genes were repeated in the two IRs (Table 1). These genes included 79 protein-coding genes, 30 tRNA genes, and four rRNA genes (Table 2). Notably, the *rps12* gene was annotated to be trans-spliced with the 3' end duplicated in IRa and IRb, and the single 5' end exon located in LSC. By comparison, the six cp genomes are uniform in gene

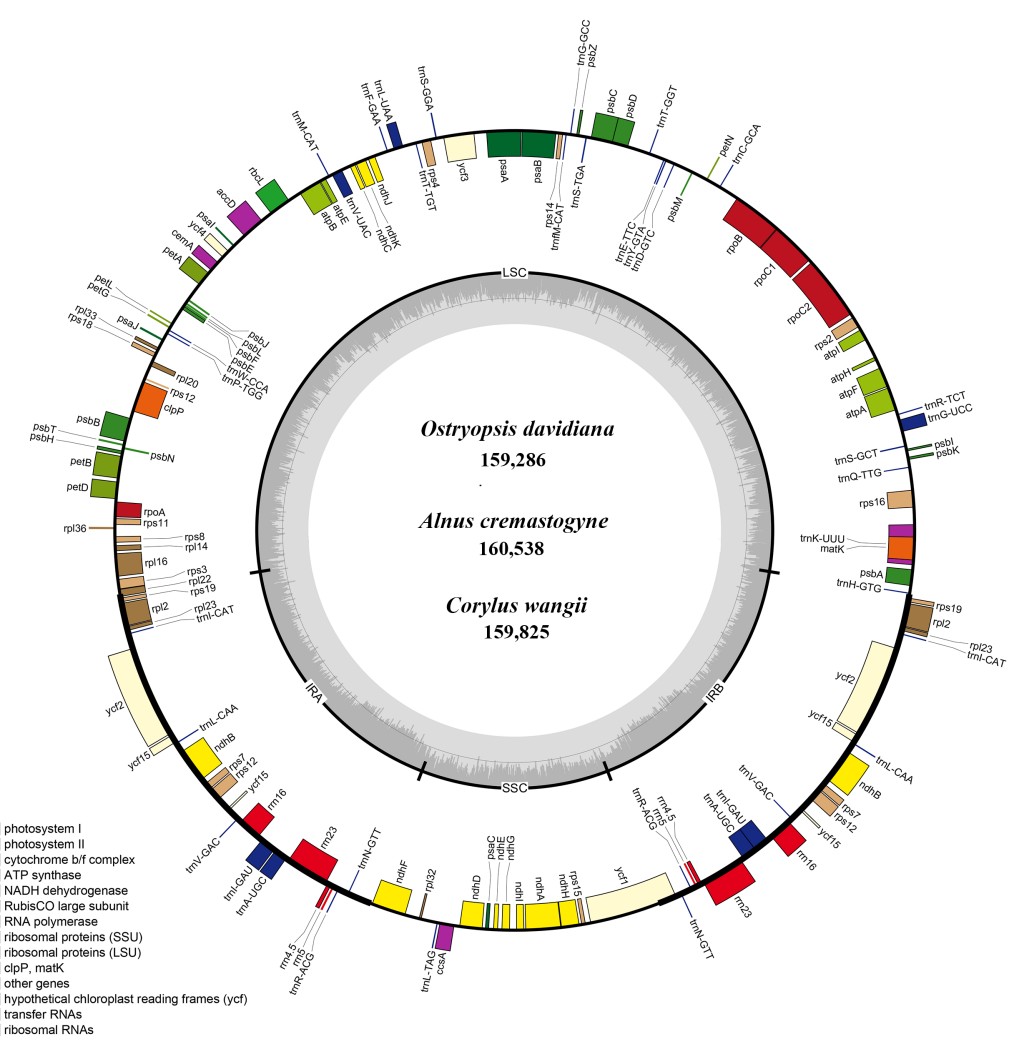

**Figure 1** **The genome maps of three Betulaceae chloroplast genomes.** The genes outside and inside of the circle are transcribed in the counterclockwise and clockwise directions, respectively. Different colors indicate the genes belonging to different functional groups. The thicknesses denote the extent of IRs (IRa and IRb) that separate the cp genomes into LSC and SSC regions.

**Table 1** **Comparison of the chloroplast genome organization among six Betulaceae species.**

| Taxon | Size (bp) | LSC (bp) | SSC (bp) | IR (bp) | Total genes | Protein coding genes | tRNA genes | rRNA genes | GC content (%) |
|---|---|---|---|---|---|---|---|---|---|
| *Ostryopsis davidiana* | 159,286 | 88,568 | 18,588 | 26,065 | 131 (18) | 86 (7) | 37 (7) | 8 (4) | 36.39 |
| *Alnus cremastogyne* | 160,538 | 89,074 | 19,094 | 26,185 | 131 (18) | 86 (7) | 37 (7) | 8 (4) | 36.68 |
| *Corylus wangii* | 159,825 | 88,743 | 18,870 | 26,106 | 131 (18) | 86 (7) | 37 (7) | 8 (4) | 36.52 |
| *Carpinus tientaiensis* | 160,104 | 89,446 | 18,598 | 26,030 | 131 (18) | 86 (7) | 37 (7) | 8 (4) | 36.38 |
| *Betula nana* | 160,579 | 89,493 | 19,018 | 26,034 | 131 (18) | 86 (7) | 37 (7) | 8 (4) | 36.07 |
| *Ostrya rehderiana* | 159,347 | 88,552 | 18,941 | 25,927 | 131 (18) | 86 (7) | 37 (7) | 8 (4) | 36.46 |
**Table 2  List of genes encoded in the chloroplast genomes of six Betulaceae species.**

| Category for genes | Group of gene | Name of gene |
|---|---|---|
| Photosynthesis related genes | Photosystem I | *psaA, psaB, psaC, psaI, psaJ* |
| | Photosystem II | *psbA, psbB, psbC, psbD, psbE, psbF, psbH, psbI, psbJ, psbK, psbL, psbM, psbN, psbT, psbZ* |
| | Cytochrome b/f compelx | *petA, petB, petD, petG, petL, petN* |
| | ATP synthase | *atpA, atpB, atpE,* [a]*atpF, atpH, atpI* |
| | Cytochrome c synthesis | *ccsA* |
| | Assembly/stability of photosystem I | [b]*ycf3, ycf4* |
| | NADPH dehydrogenase | [a]*ndhA,* [a]*ndhB(2), ndhC, ndhD, ndhE, ndhF , ndhG, ndhH, ndhI, ndhJ,* [a]*ndhK* |
| | Rubisco | *rbcL* |
| Transcription and translation related genes | Transcription | *rpoA, rpoB,* [a]*rpoC1, rpoC2* |
| | Ribosomal proteins | *rps2, rps3, rps4, rps7(2), rps8, rps11, rps12(2), rps14,rps15, rps16, rps18, rps19(2),* [a]*rpl2(2), rpl14, rpl16, rpl20, rpl22, rpl23(2), rpl32, rpl33,rpl36* |
| RNA genes | Ribosomal RNA | *rrn5(2), rrn4.5(2), rrn16(2), rrn23(2)* |
| | Transfer RNA | *trnI-CAU(2) trnI-GAU(2) trnL-UAA trnL-CAA(2) trnL-UAG trnR-UCU trnR-ACG(2) trnA-UGC(2) trnW-CCA trnM-CAU trnV-UAC trnV-GAC(2) trnF-GAA trnT-UGU trnT-GGU trnP-UGG trnfM-CAU trnG-UCC trnG-GCC trnS-GGA trnS-UGA trnS-GCU trnD-GUC trnC-GCA trnN-GUU(2) trnE-UUC trnY-GUA trnQ-UUG trnK-UUU trnH-GUG* |
| Other genes | RNA processing | *matK* |
| | Carbon metabolism | *cemA* |
| | Fatty acid synthesis | *accD* |
| | Proteolysis | [b]*clpP* |
| | Translational initiation factor | *infA* |
| Genes of unknown function | Conserved reading frames | *ycf1, ycf2(2)* |

**Notes.**
[a]gene with one intron.
[b]gene with two introns.
(2): gene with two copies.

order, gene content, and proportion of coding and non-coding regions. Accordingly, the annotated genomes were represented by one genome map (Fig. 1). Most protein-coding genes comprised only one exon, while ten genes (*atpF, rpoC1, rpl2, ndhA, ndhB, ndhK, trnV-UAC, trnI-GAU, trnA-UGC,* and *trnL-UAA*) were found to have one intron, and two genes (*clpP* and *ycf3*) contained two introns each (Table 2). The majority of the above genes were distributed in LSC and IRs, with only one gene (*ndhA*) located in SSC.

## IR contraction and expansion

To illuminate the putative contraction and expansion of IR regions, we investigated the gene variation at the IR/SC boundary regions of the six cp genomes (Fig. 2). At the IRa/LSC junctions, the gene *rps19* of *O. davidiana* and *C. wangii* crossed the IRa/LSC border, while *rps19* and *rpl2* of *A. cremastogyne, C. tientaiensis,* and *B. nana* were located in the two

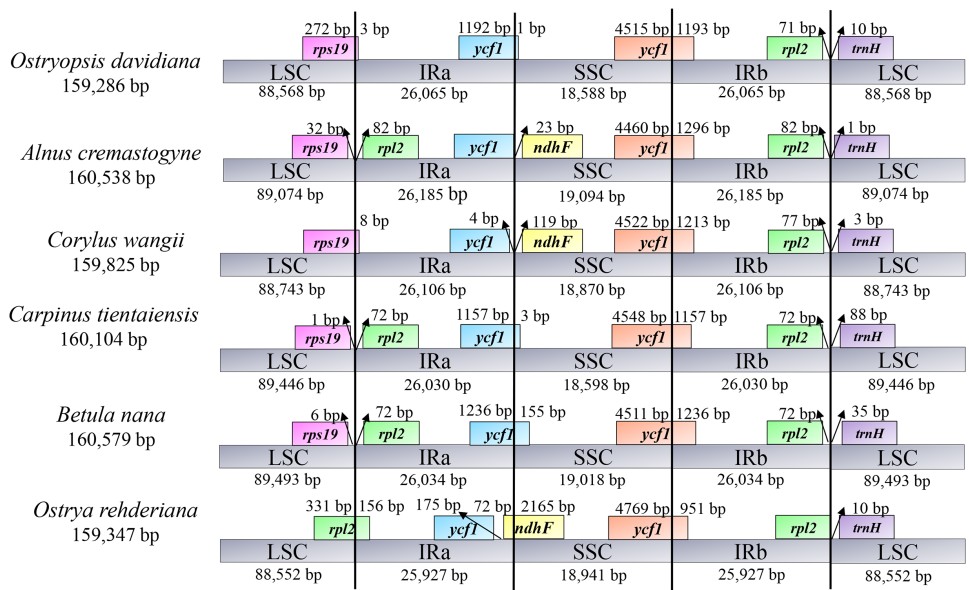

**Figure 2 Comparison of the border positions of LSC, IR and SSC among the six Betulaceae chloroplast genomes.**

sides of this border, and gene *rpl2* was created at the IRa/LSC border of *O. rehderiana*. The IRa/SSC junctions were inserted into the gene *ycf1* in three cp genomes, with 1 bp (*O. davidiana*), 3 bp (*C. tientaiensis*), and 155 bp (*B. nana*) located in the SSC region, respectively; with regard to *A. cremastogyne* and *C. wangii*, *ycf1* and *ndhF* were seated on either side of the junction; notably, the *ndhF* gene extended 72 bp into IRa region in *O. rehderiana*. In all the six cp genomes, the *ycf1* gene crossed the IRb/SSC boundary regions, resulting in the incomplete duplication of this gene in two IRs. The gene *rpl22* and *trnH-GUG* gene were distributed in the two sides of the IRb/LSC junction, with 0–82 bp for *rpl22* and 1–88 bp for *trnH-GUG* away from the junctions, respectively. IR contraction and expansion in the six Betulaceae cp genomes ultimately lead to the length variations of the four structural segments and whole genome sequences.

## Synteny analysis and divergence hotspots

In accordance with the alignment results, all the six cp genomes showed the same order and orientation of syntenic blocks (Fig. 3), indicating that Betulaceae cp genomes tend to be conserved and highly collinear, especially at the genus level. Nevertheless, a few local changes representing variable regions were still detected, with several obvious divergence fragments mainly located in SC regions, especially within the nucleotide sequences of 5,000–20,000 bp, 25,000–35,000 bp, and 135,000–145,000 bp. By contrast, the IR regions were quite conserved and no significant sequence divergence was found. Furthermore, in order to locate mutation hotspots, the variable percentages of PCG and IGS regions were calculated and analyzed (Fig. 4; Table S2). In total, cp genomes of the six Betulaceae species exhibited 7830 (5.99%) variable sites in the 130,710 sites analyzed, of which the

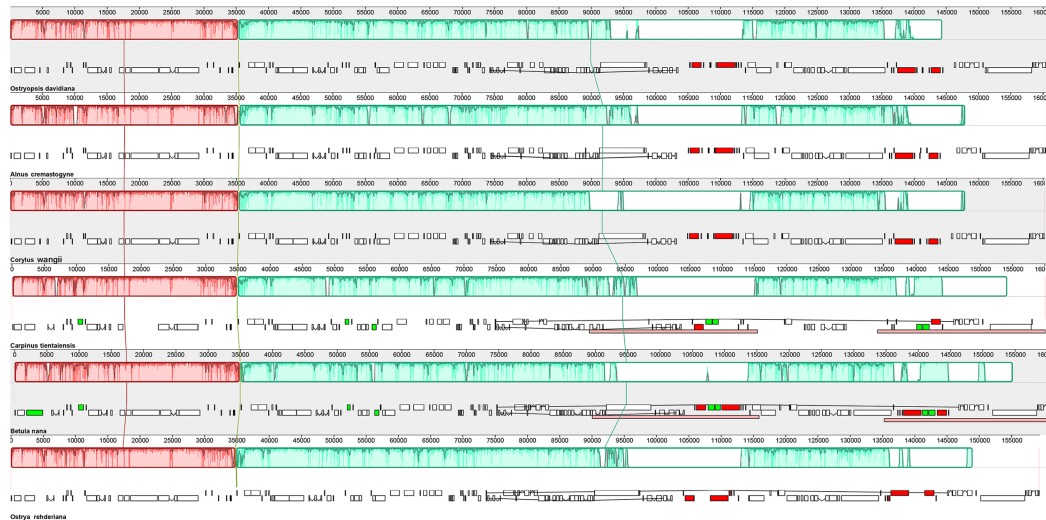

**Figure 3  Synteny and rearrangements detected in six Betulaceae chloroplast genomes using the Mauve multiple-genome alignment program.** Color plots reflect the level of sequence similarity, and lines linking blocks with the same color represent homology between two genomes. Ruler above each genome indicates nucleotide positions, and white regions indicate element specific to a genome. The above and below gene blocks are transcribed clockwise and transcribed counterclockwise, respectively.

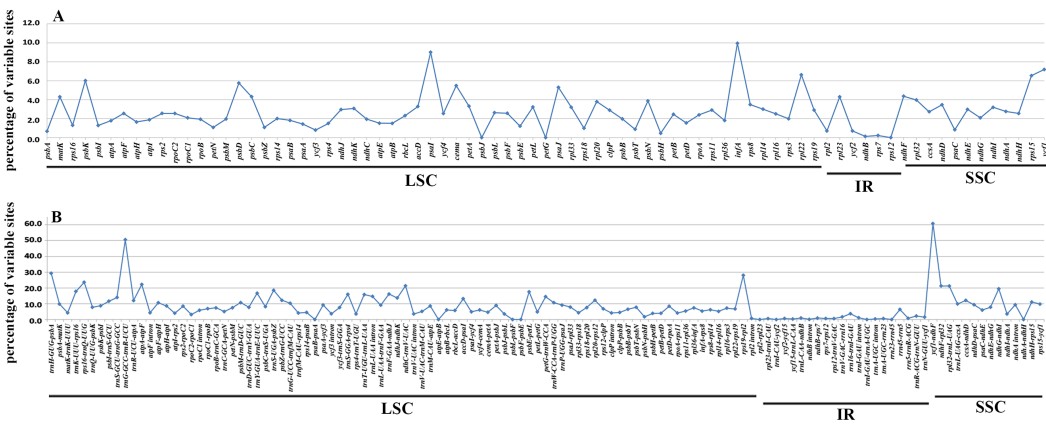

**Figure 4  Percentages of variable sites in homologous regions across the six Betulaceae chloroplast genomes.** (A) Protein-coding regions, (B) intergenic spacer regions.

average variable percentage of coding regions and intergenic spacers was 2.77% and 9.65%, respectively. The SSC region showed the highest variable percentage (9.41%), followed by the LSC region (6.56%), and then IR region (1.15%). Finally, nine hotspots (percentage of variable sites > 20%) were screened in the intergenic regions, they were: *ycf1-ndhF*, *trnG-trnR*, *trnH-psbA*, *rps19-rpl2*, *rps16-trnQ*, *atpA-atpF*, *ndhC-trnV*, *ndhF-rpl32*, and *rpl32-trnL*. Among them, five fragments were distributed in LSC, two in SSC, and two crossed the IRa/LSC and IRa/SSC boundary regions.

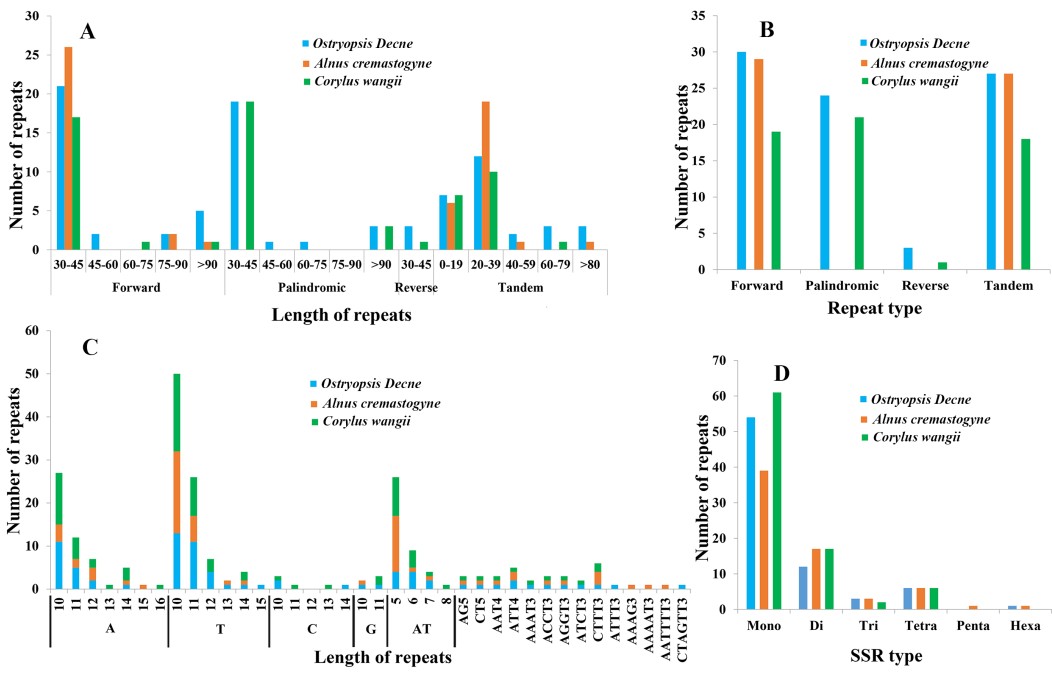

**Figure 5** **Analyses of repeated sequences and SSRs in the three Betulaceae chloroplast genomes.** (A) Frequency of repeated sequences by length, (B) frequency of four repeat types, (C) frequency of SSR motifs in different repeat class types, (D) frequency of six SSR types.

## Repeated sequences and SSRs

In the present study, four sorts of repeated sequences (forward, reverse, palindromic, and tandem) were detected in the three newly sequenced cp genomes (Figs. 5A, 5B; Tables S3, S4). Overall, 30 forward repeats, 24 palindromic repeats, three reverse repeats, and 27 tandem repeats were identified in *O. Davidiana* cp genome. In *C. wangii* cp genome, the numbers of these four repeats were 19, 21, one, and 18, respectively. By contrast, only 29 forward repeats and 27 tandem repeats were predicted in *A. cremastogyne* cp genome. The lengths of dispersed repeats (forward, palindromic, and reverse) ranged from 30 to 194 bp, with most of them centered on 30–45 bp (82.68%), while those of 45–60 bp (2.36%), 60–75 bp (1.57%), and 75–90 bp (3.15%) were relatively rare. The lengths of tandem repeats varied from 8 to 123 bp, of which a large proportion of them centered on 0–19 bp and 20–39 bp. Repeat sequences were mainly located in the non-coding regions, including IGS and introns. In addition, a few of coding genes (e.g., *ycf2*, *ycf3*, *psaA*, *atpA*, and *psaB*), tRNAs (e.g., *trnS-GGA*, *trnS-GCU*), and rRNA (e.g., *rrn16*) were also found to contain repeat structure.

Six types of SSRs (mono-, di-, tri-, tetra-, penta-, and hexa-nucleotide) were scanned within these cp genomes (Figs. 5C, 5D; Table S5). In total, 67–86 SSRs were detected, of which mono-nucleotides (especially A/T) were the most abundant, with the number ranging from 38 in *A. cremastogyne* to 56 in *C. wangii*. Di-nucleotides (especially AT) were the second most predominant, varying for 10 in *O. davidiana* and 15 in both *A. cremastogyne* and *C. wangii*. Furthermore, our data disclosed that tetra-nucleotides
which included seven sorts of sequence repeats were the most abundant SSR type, although their numbers were few. Simultaneously, a small number of tri-nucleotides were also discovered in all three cp genomes. However, only very few penta and hexa-nucleotides were detected, with one penta-nucleotide (AAAAT) and one hexa-nucleotide (AATTTT) existed in *A. cremastogyne*, and one hexa-nucleotide (CTAGTT) in *O. davidiana*. SSRs were chiefly located in non-coding regions (particularly IGS), while some coding genes (e.g., *psbI*, *rpoC2*, *rpoB*, *atpF*, and *atpB*) were also found to hold SSRs. On the whole, SSRs were unevenly scattered throughout the four structural domains of cp genomes, with most of them distributed in LSC, followed by SSC and IR.

## Phylogenetic inference

Both the ML and BI phylogenies inferred from CPG and PCG datasets displayed nearly identical topologies in identifying the taxonomic status of six genera (Fig. 6A, Fig. S1). All the nodes were moderately or highly supported. The eleven ingroup taxa were divided into two major clades, which accorded well with traditionally divided Coryloideae and Betuloideae. The subfamily Coryloideae was a large clade constituted by four genera (*Corylus*, *Ostryopsis*, *Carpinus*, and *Ostrya*), while Betuloideae consisted of the other two genera (*Betula* and *Alnus*), of which *Carpinus-Ostrya* and *Alnus-Betula* formed two stable sister subclades. The two *Juglans* species were included in outgroup. Although the intergeneric relationships revealed by IGS data were mostly consistent with that of CPG and PCG datasets, visible incongruence on the phylogenetic position of *Ostryopsis* was still observed. The CPG and PCG phylogenies placed *Ostryopsis* basal to the *Carpinus-Ostrya* subclade (Fig. 6A, Fig. S1), while the IGS phylogeny supported it sister to *Corylus* (Fig. 6B).

## Divergence times and ancestral areas

The tree topology inferred from the molecular dating analysis (Fig. 7) was consistent with those recovered from CPG and PCG datasets (Fig. 6A, Fig. S1). All the nodes in the tree were highly supported with a posterior probability of 1. The estimated divergence time and 95% highest posterior density (HPD) were displayed on the branches. Betuloideae and Coryloideae diverged in the late Cretaceous (~70.49 Mya, 95% HPD = 66.62–74.29 Mya), as their most probable time of origin. The divergence of Betuloideae into *Betula* and *Alnus* occurred in the middle Paleocene (~61.76 Mya, 95% HPD = 49.77–70.97 Mya). The MRCA of Coryloideae and the split of *Corylus* occurred in the early Eocene (47.93 Mya, 95% HPD = 46.95–48.91 Mya). The divergence time between the genus *Ostryopsis* and the sister group of *Ostrya-Carpinus* was around 44.63 Mya (95% HPD=40.11–47.93 Mya), which was a little later than *Corylus* (~3 Mya). The diversification of the sister subclade (*Ostrya* and *Carpinus*) was suggested to be 26.73 Mya (95% HPD = 15.09–39.44 Mya) in the late Oligocene. BBM analysis suggests that intercontinental dispersals played important roles in the biogeographic history of Betulaceae (Fig. 8). However, the origin area of the six extant genera was unclear because of the insufficient species sampling, and uncertainty of its sister group in previous studies. In spite of this, we identified three major distribution areas: East Asia (A), Europe (B), and North America (C) which were speculated to break away and drift from the old Laurasia in the Paleozoic (~57–23 Mya). The extant species

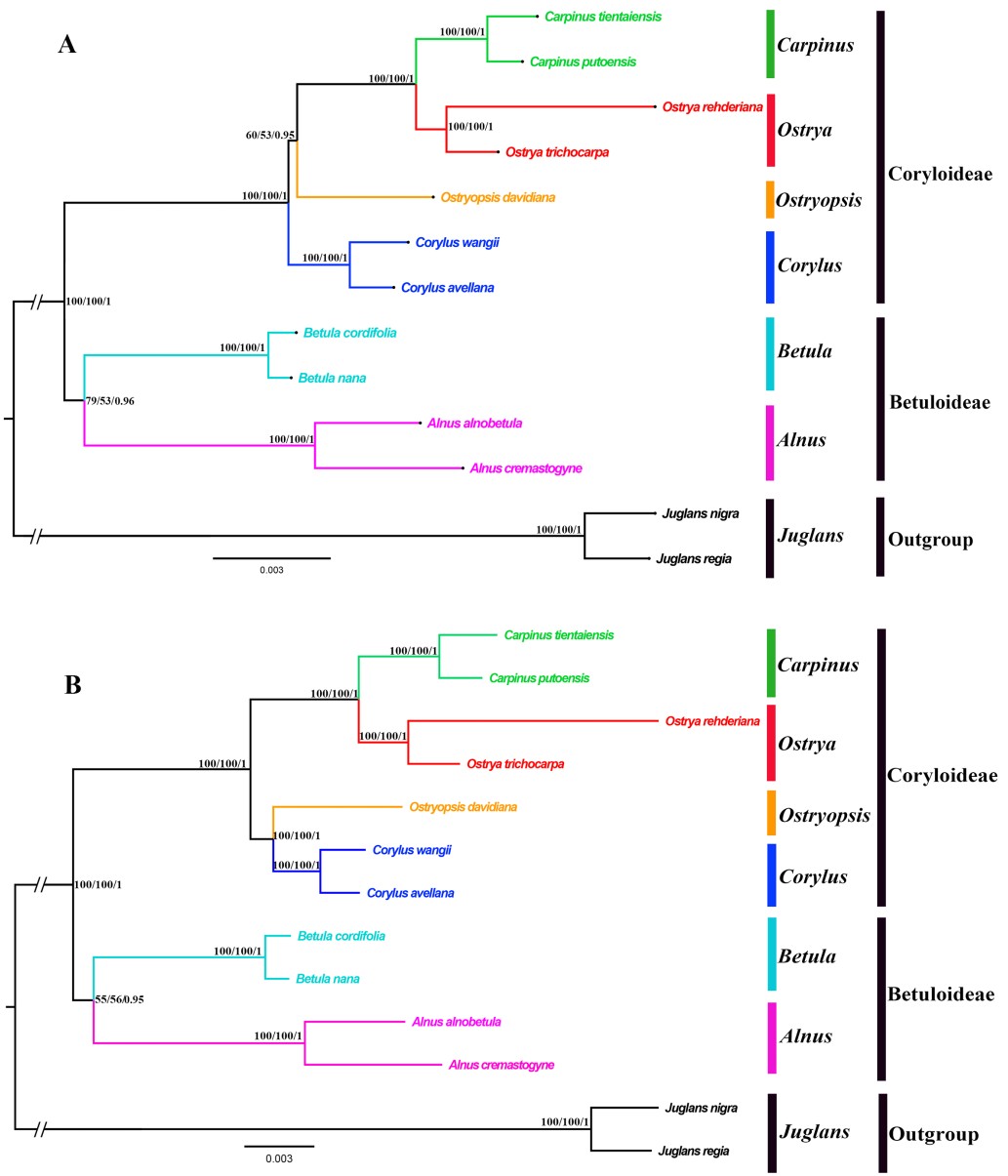

**Figure 6  Phylogenetic trees of Betulaceae as inferred from two data partitions using ML and BI methods.** (A) complete cp genome sequences (CPG), (B) intergenic spacer regions (IGS). Support values of ML-SH-Alrt, ML-UFBoot and BI-PP are successively listed above the branches (SH-aLRT/*UFBoot /PP*).

of three genera (*Alnus*, *Ostrya*, and *Carpinus*) that exist in Central America (D) and South America (E) may have originated in North America (C) and traveled across the Isthmus of Panama to South America. While a few *Alnus* species have spanned the island chains constituted by Balkan Peninsula, Southern Turkey, and Italy into North Africa (F).

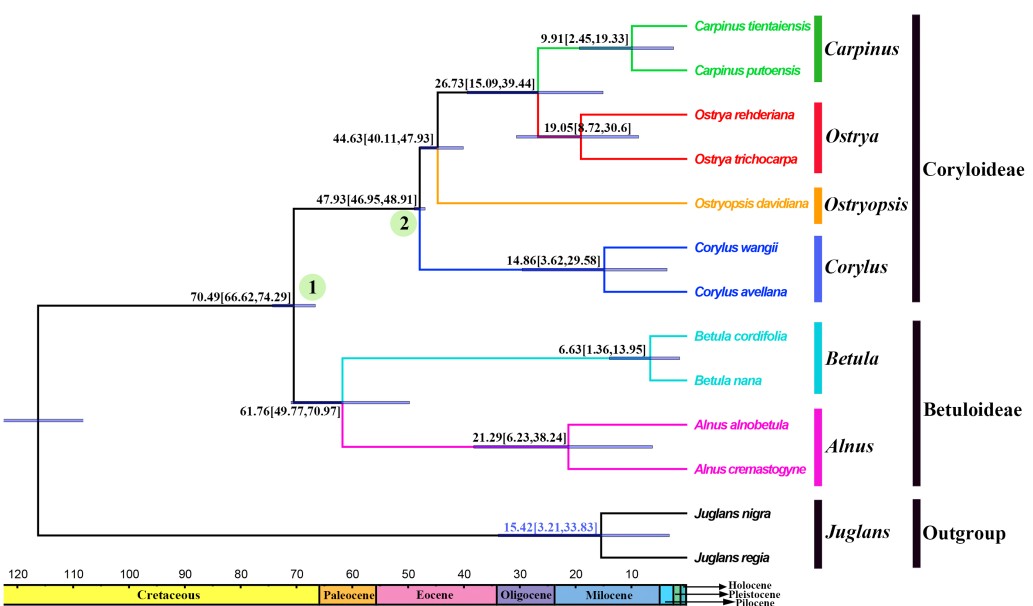

**Figure 7** **Fossil-calibrated phylogeny generated by BEAST using an uncorrelated relaxed clock.** Blue bars on the nodes indicate 95% highest posterior density. Divergence time of clades and subclades are displayed on the branches.

## DISCUSSION

In the research, we characterized the cp genomes of three Betulaceae species, identified SSRs, repeated sequences, divergence hotspots throughout these genomes, and performed phylogenetic analyses by integrating closely related cp genomes. Correspondingly, these findings also provide an opportunity to explore the divergence history of Betulaceae species. Our research has laid the foundation for future studies on the evolution of *Ostryopsis*, *Alnus*, and *Corylus*, as well as the molecular identification of Betulaceae species.

### Chloroplast genome variation and evolution

The cp genomes of most angiosperms are validated to contain approximately 130 genes, of which about 20 genes have two copies in two IRs, leaving the rest 110 being unique genes (*Mader et al., 2018*; *Hu, Woeste & Zhao, 2017*; *Xu et al., 2017*; *Yang et al., 2018*). Our annotations are similar to those reported above. Comparative analysis indicates that Betulaceae cp genomes possess a set of 113 unique genes, including 79 protein-coding genes, 30 tRNAs, and 4 rRNAs (Table 1). The differences of cp genome size (varying from 159,347 to 160,579 bp) reflect the genome variation of Betulaceae species. In general, this phenomenon may arise from the contraction and expansion of IR regions, and has been reported in many plant cp genomes (*Zhang et al., 2017*; *Lu, Li & Qiu, 2017*). Similarly, despite the conservative property of Betulaceae cp genomes, changes in the IR/SC junctions were also observed, indicating the cp genome variation and evolution to some extent.

It has been proved that comparative genomics contributes to the development of divergence hotspots which can be applied for species identification (*Ahmed et al., 2013*)

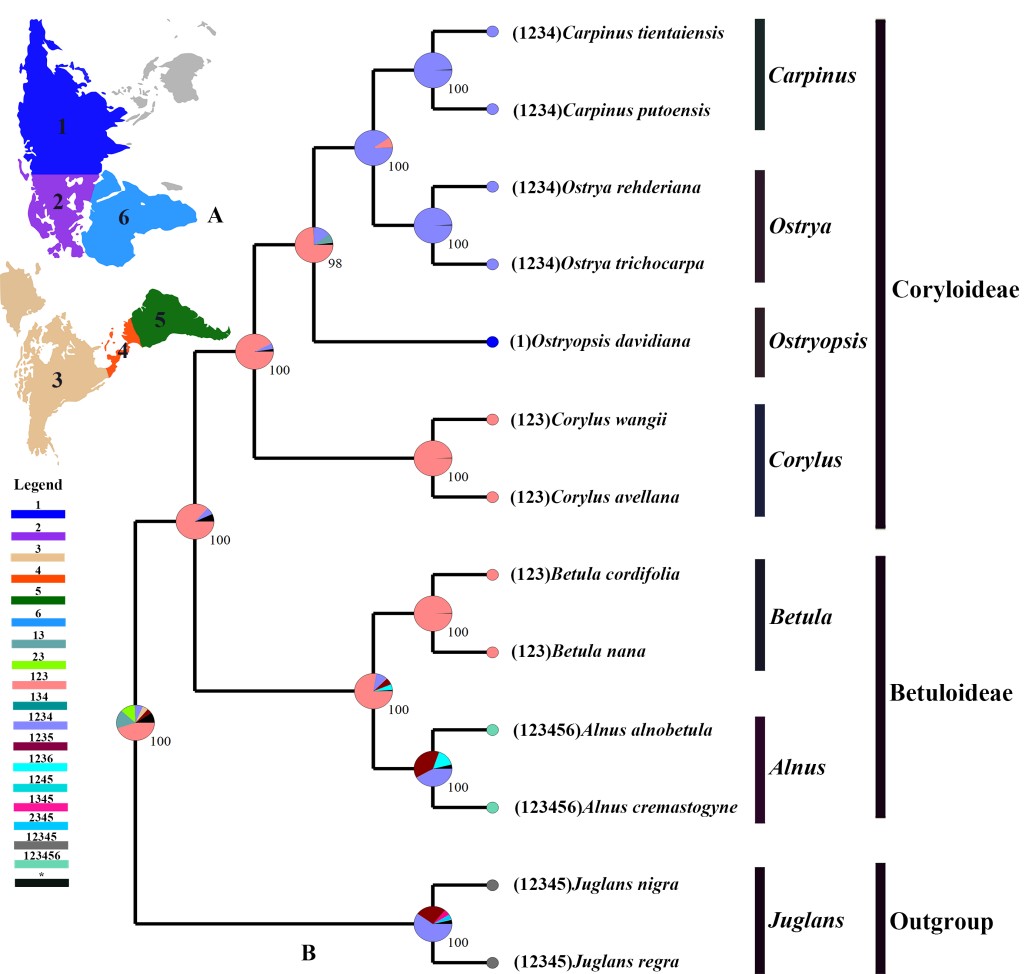

**Figure 8** **Ancestral area reconstruction based on the BBM method in RASP.** (A) The insert map shows the contemporary distribution of Betulaceae species, covering six major floristic divisions (1–6). (B) Pie charts on each node of the tree indicate marginal probabilities for each alternative ancestral area. Numbers and colors in the legend refer to extant and possible ancestral areas, and combinations of these.

and phylogenetic studies of different levels (*Downie & Jansen, 2015*; *Shaw et al., 2014*). Previous studies have confirmed that several protein-coding genes of cp genomes were very efficient in resolving the phylogenetic relationships of some complex plant taxa, e.g., *petB*, *rps16*, *psaI*, *rps11* and *rpoA* in *Notopterygium* species (*Yang et al., 2017*), and *ycf1* gene in *Anemopaegma* species (*Firetti et al., 2017*). Furthermore, more studies reveal that the intergenic spacer regions had higher resolution in species delimitation of related plant taxa, e.g., *psaC-ndhE*, *rpoB-trnC*, *clpP-psbB*, *rpl32-trnL*, *trnT-psbD*, and *ccsA-ndhD* had significant genetic divergence among *Phalaenopsis* species (*Shaw et al., 2014*), and *petD-rpoA*, *trnT-trnL*, *trnG-trnM*, *ycf4-cemA*, and *rpl32-trnL* could be used to identify Veroniceae species (*Choi, Chung & Park, 2016*). In this research, both variable percentage and synteny analysis of Betulaceae cp genomes indicate that IGS had higher variation than PCG, from which nine intergenic spacer fragments are identified as divergence hotspots

(percentage of variable sites > 20%) (Fig. 4B; Table S2). Two protein-coding genes (*psaI* and *infA*) show higher variable rate (percentage of variable sites > 8%) than other genes (Fig. 4A; Table S2). Despite of this, the practical application of these hotspots remains to be verified using methods of population genetics.

Repeated sequences play key roles in cp genome rearrangement, divergence, and evolution, while SSRs are extensively applied in population genetics and molecular identification (*Weng et al., 2013*; *Xue, Wang & Zhou, 2012*; *Guisinger et al., 2010*). The presence of repeated sequences in cp genomes, especially in IGS, has been discovered in many known angiosperm lineages (*Xue, Wang & Zhou, 2012*; *Xu et al., 2017*; *Yang et al., 2017*). Similarly, we identify four sorts of repeated sequences and six types of SSRs that distribute widely in IGS of the three Betulaceae cp genomes. Moreover, the three cp genomes present obvious differences in both the distribution pattern and number of dispersed repeats; however, no significant differences are observed in tandem repeats (Fig. 5; Tables S3, S4). Notably, *C. wangii* cp genome contains the most abundant SSRs among the three species although its genome size is the smallest, which can be used as the unique identification for this species. Furthermore, these cpSSRs are rich in thymine or adenine repeats, but rarely contains guanine or cytosine repeats. Similar findings are also discovered in the cpSSRs of other plant taxa such as *Scutellaria* (*Jiang et al., 2017*), *Salvia* (*Qian et al., 2013*) and *Juglans* (*Hu, Woeste & Zhao, 2017*). These newly developed repeats and SSRs would be helpful for detecting genetic polymorphisms at population level and assessing distantly related evolutionary relationships within Betulaceae.

## Evolutionary relationships within Betulaceae

Betulaceae are a monophyletic family in the order Fagales and are traditionally divided into two main clades, treated as two subfamilies (Coryloideae and Betuloideae) (*Forest et al., 2005*; *Chen, Manchester & Sun, 1999*). However, the intergeneric relationships within this family are still not clearly resolved because previous phylogenetic conclusions in Betulaceae were inferred either based on morphological characters (*Stone, 1973*; *Abbe, 1974*) or several molecular fragments such as chloroplast *matK* gene (*Kato et al., 1998*), *rbcL* gene (*Bousquet, Strauss & Li, 1992*), as well as nuclear ITS sequences (*Chen, Manchester & Sun, 1999*). Compared with those morphological markers and single loci, complete cp genome undoubtedly have more advantages to resolve the phylogenetic problems of Betulaceae lineages. In this research, all the phylogenies inferred from the three datasets (CPG, PCG, and IGS) are in favor of the division of Coryloideae and Betuloideae, as well as the same genus composition to previous studies within each subfamily (Fig. 6, Fig. S1). Nevertheless, two different topologies occur within Coryloideae, with the most apparent discrepancy consisting in the phylogenetic position of *Ostryopsis*. The CPG and PCG datasets reveal a close affinity between *Ostryopsis* and the *Carpinus-Ostrya* subclade, while *Corylus* formed sister group to the three genera (Fig. 6A, Fig. S1). This kind of generic relationship is in accordance with that inferred from ITS and *rbc* L phylogenies (*Chen, Manchester & Sun, 1999*; *Bousquet, Strauss & Li, 1992*). By contrast, the IGS dataset supports a sisterhood between *Ostryopsis* and *Corylus* (Fig. 6B), which is identical with the phylogenetic inference of *mat* K sequences (*Kato et al., 1998*). We infer that the incongruence among different

datasets may probably be related with various evolutionary rates of different nucleotide regions, which deserves our further validation.

## Divergence history and biogeography

Betulaceae are suggested to have originated in the late Cretaceous (∼70 Mya) in central China of East Asia (*Christenhusz & Byng, 2016*; *Soltis et al., 2011*). Due to the proximity of the Tethys Sea, this region at that time may have belonged to the Mediterranean climate which covered parts of present-day Xinjiang and Tibet until the early Tertiary period. This biogeographic origin is favored by the fact that all the six extant genera and nearly one third of species in Betulaceae are native to this region. Our molecular dating analysis supported Betulaceae to be originated at the end of Cretaceous (∼70.49 Mya), which is very close to the above results. Due to the limited representative species and outgroup used in our analysis, ancestral area reconstruction does not designate an exact origin region. However, we can confirm that ancestors of extant Betulaceae species were once extensively distributed in Laurasia that covered the present-day Asia, Europe, and North America, from which some species have dispersed into Central America, South America, and North Africa through different island chains. Those intercontinental dispersals are also validated from the biogeography of other angiosperms (*Morley, 2003*; *Sanmartin & Ronquist, 2004*). On basis of some morphological characters such as three-flowered cymules, bisexual inflorescences, and staminate flowers, the genus *Alnus* is suggested to be the earliest to split from the ancestor of the Betulaceae because it preserves certain primitive and unique characters of this family (*Chen, Manchester & Sun, 1999*). *Betula* appears in some aspects to be transitional between Coryloideae and *Alnus*, with characters of fruit, cymule, and inflorescence being similar or identical to those of *Alnus*, while other features are akin to those of Coryloideae. Similarly, *Corylus* is assigned to be intermediate between Betuloideae and Coryloideae because it possesses some common characters shared with *Betula* and *Alnus*, as well as the characters peculiar to *Ostryopsis*, *Carpinus*, and *Ostrya*. Our molecular dating analysis indicates that the divergence order of the six genera is *Alnus*, *Betula*, *Corylus*, *Ostryopsis*, *Ostrya*, and *Carpinus* in sequence, which corresponds consistently with the morphological evolution. On the basis of above analyses, detailed taxon sampling needs to be carried out so as to obtain a biogeographic history of Betulaceae on a large-scale.

## CONCLUSIONS

Betulaceae cp genomes are highly conserved in genome organization, gene order, and gene content, indicating low-level genome variation. Sequence divergence in SC is higher than IR, and IGS have higher variation than PCG. Nine IGS regions (*ycf1-ndhF*, *trnG-trnR*, *trnH-psbA*, *rps19-rpl2*, *rps16-trnQ*, *atpA-atpF*, *ndhC-trnV*, *ndhF-rpl32*, and *rpl32-trnL*) may be applied in future population genetics and phylogenetic studies of Betulaceae. The phylogenetic inference supports the division of Betulaceae into two subfamilies: Coryloideae and Betuloideae. *Ostryopsis* is a transitional genus between *Corylus* and *Carpinus-Ostrya*. *Alnus* and *Betula* of the Betuloideae differentiate earlier than *Corylus*, *Ostryopsis*, *Ostrya*, and *Carpinus* of the Coryloideae. More detailed taxon sampling will contribute to the comprehensive phylogenetic study.

## ACKNOWLEDGEMENTS

We thank Dr. Xinhui Zou and Wenpan Dong of Institute of botany, the Chinese Academy of Sciences for the help in experiment design and genome sequencing. We also thank Xinming He (Yunnan, China) for providing samples.

### Funding

This study was supported by the Special Fund for Basic Scientific Research Business of Central Public Research Institutes (Grant Nos. CAFYBB2018SY011 and CAFYBB2017ZA004-9) and the National Natural Science Foundation of China (Grant No. 31500555). The funders had no role in study design, data collection and analysis, decision to publish, or preparation of the manuscript.

### Grant Disclosures

The following grant information was disclosed by the authors:
Basic Scientific Research Business of Central Public Research Institutes: CAFYBB2018SY011, CAFYBB2017ZA004-9.
National Natural Science Foundation of China: 31500555.

### Competing Interests

The authors declare there are no competing interests.

### Author Contributions

- Zhen Yang and Tiantian Zhao conceived and designed the experiments, performed the experiments, analyzed the data, contributed reagents/materials/analysis tools, prepared figures and/or tables, authored or reviewed drafts of the paper, approved the final draft.
- Guixi Wang conceived and designed the experiments, contributed reagents/materials/-analysis tools, prepared figures and/or tables, authored or reviewed drafts of the paper, approved the final draft.
- Qinghua Ma, Wenxu Ma and Lisong Liang performed the experiments, contributed reagents/materials/analysis tools, approved the final draft.

### DNA Deposition

The following information was supplied regarding the deposition of DNA sequences:
The three newly sequenced chloroplast genome sequences described here are accessible via FigShare: Yang, Zhen (2018): Three Betulaceae chloroplast genomes. figshare. Fileset.
10.6084/m9.figshare.7199816.v1.

### Data Availability

The three newly sequenced cp genomes have been submitted to GenBank under accession numbers (MH628451 for *Ostryopsis Davidiana*, MH628454 for *Corylus wangii*, and MH628453 for *Alnus cremastogyne*).

## Supplemental Information

Supplemental information for this article can be found online at http://dx.doi.org/10.7717/peerj.6320#supplemental-information.

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
