# Peer review of "The complete chloroplast genomes of three Betulaceae species: implications for molecular phylogeny and historical biogeography"

_PeerJ, doi:10.7717/peerj.6320_

## Round 0.1 · original submission · Major Revisions

Please address all the questions raised by the reviewers

·

Basic reporting

Authors sequenced and analyzed the chloroplast genome of three Betulaceae species. Authors identified divergence hotspots throughout these genomes including repeats which can be used as markers for Betulaceae species.

Introduction is written in a complex way and sufficient background is not provided to understand the significance of the work. It should be rewritten with sufficient background. For example, lines 54-55, “Opinions differ, however, relating to the intergeneric relationships and genera circumscription as ranking into higher taxonomic level” is not clear what authors want to say here.

In figure 6, what is the criteria for the selecting node in phylogenetic analysis? It is not clear how the phylogenetic tree was generated? How many sequences were used and the basis of the selection ?

Experimental design

Methods should be described in detailed. All the previously described methods/computational programs should be briefly described. For example, “BEAST v2.4.8 software”, authors should explain why this software was used and what are parameters used in this software.

What is the average read length in the sequencing data? How much error rate is present in the analyzed sequences? How this error rate affected the SSRs in the analyzed data and the phylogenetic analysis?

Overall detailed description is required to understand all the parameters used in all analysis.

Validity of the findings

Conclusions are well stated.

Additional comments

Authors sequenced and analyzed the chloroplast genome of three Betulaceae species. Authors identified divergence hotspots throughout these genomes including repeats which can be used as markers for Betulaceae species. Manuscript is written well however authors need to address following issues.

1. Authors should described the parameters used in the phylogenetic analysis.

2. Figure 3 is difficult to follow. Authors can add descriptions in figure legend.

3. Figure 4 shows the variability in protein coding region or intergenic spacer regions. How this variability correlated with numbers given in Table 1?

Reviewer 2 ·

Basic reporting

In this work, the authors sequenced chloroplast genome of three Betulaceae species, and performed comparative genomics and phylogenetic analysis on the sequenced genomes along with some of the previously published chloroplast genomes of other Betulaceae family members to highlight genome variations, identify molecular markers, and resolve phylogenetic relationship and divergence history of Betulaceae family. The manuscript is well written, clearly explained, and there are no significant technical concerns.

Experimental design

The aims of the study are well-defined and the methods used are appropriate and sufficiently detailed. Although some clarifications are warranted as noted in general comments.

Validity of the findings

Overall, conclusions are well supported by the data.

Additional comments

1) While the phylogenetic trees constructed using the CPG/IGS dataset (Fig 6A and S1) differed from that constructed using PCG dataset (Fig 6B) with respect to Ostryopsis position, it is interesting to note that the topology obtained with CPG dataset (Fig S1) also differs from the topology obtained using CPG dataset in Fig 6 of Yang et al, 2018 (PMID: 30038632), wherein the Ostryopsis position in the CPG dataset topology is similar to the PCG dataset topology obtained here (Fig 6B). Can the authors comment on incongruence between the two topologies obtained using CPG datasets? Would the topology obtained using CPG dataset be different if the authors also include the other chloroplast sequences from Yang et al, 2018 (PMID: 30038632) (for example including the Ostryopsis Decaisne and/or other Corylus sequences)?
2) On line 92-93, authors should provide the reason for selecting these 3 species in particular for chloroplast genome sequencing.
3) On line 120, “…. were checked by aligning with homologous genes of other cp genomes…”, authors should clarify what other chloroplast genomes were used?
4) On line 127-128, accession numbers should be provided for the chloroplast genomes of Carpinus tientaiensis, Betula nana, and Ostrya rehderiana.
5) Authors should provide the accession number of all the chloroplast genome sequences used to construct phylogenetic tree.
6) Some of the papers cited in the manuscript are missing from the References list. For example, Furlow 1990 on line 46, Heywood, 1993 on line 58 etc.
7) Abbreviations are used without first defining it. For example, ITS on line 67.

---

## Round 0.2 · accepted · Accept

Authors have addressed all the questions raised by both the reviewers and incorporated all the suggested changes. Manuscript is now suitable for publication.

# ·

Basic reporting

Authors sequenced and analyzed the chloroplast genome of three Betulaceae species. Authors identified divergence hotspots throughout these genomes including repeats which can be used as markers for Betulaceae species. In revised manuscript, all the comments have been incorporated and manuscript has been improved.

Experimental design

In revised manuscript, authors have included details of all the experimental analysis.

Validity of the findings

Current manuscript show the detailed analysis of Betulaceae species and identified variations in genome that can be used as markers to identified the species.

Additional comments

Overall, manuscript has been improved.

Reviewer 2 ·

Basic reporting

no comment

Experimental design

no comment

Validity of the findings

no comment

Additional comments

Authors have addressed all my concerns and manuscript is now suitable for publication.